# Soybean Molasses in Animal Nutrition

**DOI:** 10.3390/ani11020514

**Published:** 2021-02-16

**Authors:** Sladjana Rakita, Vojislav Banjac, Olivera Djuragic, Federica Cheli, Luciano Pinotti

**Affiliations:** 1Institute of Food Technology, University of Novi Sad, Bulevar cara Lazara 1, 21000 Novi Sad, Serbia; vojislav.banjac@fins.uns.ac.rs (V.B.); olivera.djuragic@fins.uns.ac.rs (O.D.); 2Department of Health, Animal Science and Food Safety, VESPA, University of Milan, 20134 Milano, Italy; federica.cheli@unimi.it (F.C.); luciano.pinotti@unimi.it (L.P.); 3CRC I-WE (Coordinating Research Centre: Innovation for Well-Being and Environment), University of Milan, 20134 Milan, Italy

**Keywords:** soybean molasses, food industry by-products, sustainable animal nutrition, alternative feed ingredients, ruminants, pigs

## Abstract

**Simple Summary:**

Soybean molasses is a by-product of the soybean processing industry that is accumulated in large quantities and usually disposed of like liquid manure or in landfills, thus causing severe ecological problems. At the same time, soybean molasses has a promising potential to be included regularly in animal diets because of its high nutritive value and functional properties. It is rich in sugars and is a cheap energy source for animals compared to other energy-rich feed ingredients. This paper reviews current knowledge on the valorization of soybean molasses as an alternative feed ingredient focusing on its composition and application in animal nutrition in general.

**Abstract:**

Concerning the increasing global demand for food and accumulation of huge amounts of biomass waste from the agro-food industry whose manipulation is usually inadequate, the potential of livestock to convert by-products as alternative feed ingredients into valuable proteins has been proposed as an outstanding option. Soybean molasses present a by-product of soybean protein concentrate production with low commercial cost but high nutritive and functional value. It is a rich source of soluble carbohydrates in the form of sugars and soybean phytochemicals. Therefore, this paper provides a review of published works about the production of soybean molasses, chemical composition, and nutritive value. In addition, the possibility of the application of soybean molasses in animal nutrition as a pelleting aid and functional feed ingredient is also discussed. Special attention is devoted to the influence of the inclusion of soybean molasses in the diets for ruminants, non-ruminants, and aquaculture on animal performance and health.

## 1. Introduction

Global demand for food is constantly increasing and it is driven by both population growth and a rise in consumption in developing regions due to increased living standards. As the global resources are limited, changes in approach are required, i.e., a shift from linear to more efficient circular agro-food systems. Maximizing circularity and cascading effects between crops, livestock, and the food industry will help to produce animal products with fewer edible resources and less environmental impact thus maximizing the global efficiency of the food systems (Figure 1). To achieve this goal, non-human-edible biomass from the food production process can be valorized as by-products of the food chain. Using the potential of livestock to convert inedible products/alternative feed to valuable proteins is necessary for optimal utilization of biomass components across industries [1].

According to Food and Agriculture Organization FAO [2], huge quantities of biomass waste are generated as by-products of food and agricultural industries, accounting for approximately one-third of the global agricultural production. International non-governmental organizations are highlighting an urgent need for waste minimization and efficient valorization of valuable resources [3,4]. Furthermore, livestock production is rapidly growing and it approximately accounts globally for 40% of the total agricultural activity and for more than 50% in developed countries. It is predicted that the global demand for animal products in developing countries will substantially increase by 2030 [5]. From that point of view, finding suitable alternative feed ingredients in animal nutrition to produce animal products has become a very challenging mission [6]. Based on the abovementioned, the conversion of agricultural by-products, crop residues, and edible biomass into ingredients for animal feed has been regarded to be an efficient tool that may reduce food-feed competition, decrease animal feed costs, alleviate the environmental impacts of livestock production, and diminish the dependence on grains widely used in human consumption [7,8,9,10,11,12]. This concept points out the principles of waste reduction, nutrients recycling, efficient reuse of resources, and development of production process adjusted to unique local conditions [11,13]. Tight cooperation between nearby food producers and local feed factories and farmers has been established over a long period and still continues [11,14]. This is important because animal feeds have become a growingly critical part of the integrated food chain. Since the livestock sector demands innovation to secure sustainable feed and food security, the concept of recycling is an alluring approach in many ways and should thus be implemented globally [15].

Soybean molasses is a relatively new investigated by-product of soybean processing generated in large quantities and with, currently, low commercial cost [16]. This by-product has been mainly used as a raw material for fermentation in the production of bioethanol, lactic acid, and various solvents (acetone, butanol) [17,18]. It was reported that leading soybean processors in Brazil generate a substantial quantity of soybean molasses per day (approximately 220 tons) [19]. Since soybean molasses present agro-industrial waste with a high amount of organic substances, it has been discarded in landfills or disposed of like liquid manure up to now. Consequently, organic compounds from soybean molasses contaminate groundwater and pollute the air, thus causing environmental issues [19]. Therefore, adequate valorization of soybean molasses, i.e., application in animal nutrition can reduce inappropriate disposal of molasses that induces environmental problems on one hand, and contributes to the reduction of feed costs and animal products production on the other hand [19]. The potential of soybean molasses as a feed ingredient has not been thoroughly explored yet. However, the incorporation of soybean molasses in animal diets in accordance with other nutritional requirements can easily enhance and enrich conventional animal diets.

This review aims to present soybean molasses as an alternative feed ingredient with special emphasis on its composition, application in animal diets as a pelleting aid, and source of energy and functional compounds. 

## 2. Soybean Molasses Production

Soy molasses is a brown-colored viscous syrup with a characteristic bittersweet taste [20]. It is obtained from defatted soybeans by the alcohol extraction, and it is a by-product of aqueous alcohol soybean protein concentrate production. After removing the oil from the crushed soybeans, molasses is obtained by the industrial extraction of defatted soybean flakes (white flakes) or soybean meal, by use of warm diluted ethanol (60–70%) or diluted iso-propanol to remove carbohydrates and concentrate the protein. The choice of alcohol type for extraction depends on the availability and prices of extraction liquid. To the best of our knowledge, there is no publication that shows the influence of type of alcohol used for the extraction on nutritional quality of soybean molasses. After extraction, alcohol and some water are removed from the liquid fraction by evaporation, distillation, stripping with steam, elution, column separation, membrane separation, or other techniques. The end product, syrup, or soy molasses is virtually alcohol-free, with the desired moisture content [18,21]. The production of soybean molasses in the industrial processing of soybean is shown in Figure 2.

The yield of soybean molasses is approximately 19% counting on the soybeans, or 25% counting on the defatted soybean meal, which is considerably high. Moreover, due to the increase in global demand for valuable proteins, amounts of produced soybeans are constantly increasing, consequently followed by an increase in quantities of soybean molasses. Therefore, for optimal usage of the soybeans, valorization of all by- and co-products is justified from multiple aspects.

It is possible to produce modified soybean molasses with reduced sugar content by partial or complete elimination of sugars from the molasses. The following methods can be used to remove sugars from soybean molasses: microbial fermentation, enzymatic and chemical treatments, gel filtration, column separation, membrane separation, acid precipitation to precipitate the major non-sugar components and removal of the soluble components, mainly the sugars, by centrifugation, settling, or decantation [23].

## 3. Composition of Soybean Molasses

The composition of soybean molasses can be variable and it highly depends on the soybean variety, growing conditions, location, and year of cultivation [20]. Soybean molasses is mainly composed of carbohydrates, proteins, and lipids. The typical composition of soybean molasses is shown in Table 1. 

According to Chajuss [20], the chemical composition of soybean molasses is similar to that from beet molasses and cane molasses. It was reported that soybean molasses contains a high content of dry matter (82.5%), the crude protein content of 5–7%, and the mineral content of 3–7% [20]. On the other hand, significant differences in terms of the chemical composition of soybean molasses in relation to beet or sugar cane molasses can be observed in the research of Long and Ribbons [18]. They stated significantly lower dry matter content in soybean molasses (53.1%), but higher crude protein content (11.7%) and significantly higher mineral content (21.9%). In addition, Loman and Lu-Kwang [26] reported higher protein content (8–12%), lower ash content (5–7%), and significantly higher fat content (15–20%) in soybean molasses. Romão et al. [27] also reported fat content in molasses to be approximately 15% on a dry basis. The major components of soybean molasses are carbohydrates accounting for 58–65% dry matter basis. The following carbohydrates can be found in molasses: oligosaccharides (mainly stachyose and raffinose), disaccharides (saccharose), and smaller proportions of monosaccharides (fructose and glucose). Stachyose and saccharose are the most abundant carbohydrates in soybean molasses accounting for up to 34.2% and 32%, respectively [18,20]. Soybean molasses contains less amount of raffinose (up to 9%); however, Long and Ribbons [18] reported a significant amount of this carbohydrate in molasses to be 25.5%. Monosaccharides fructose and glucose are present in molasses in smaller amounts. Among minerals, soybean molasses is rich in potassium accounting for 7.2% dry matter basis, while contains a lower amount of magnesium, phosphorus, chlorine, calcium, and sodium (0.7%, 0.58%, 0.44%, 0.31%, and 0.26%, respectively [28]. The data presented in Table 1 indicate that soybean molasses varies widely in chemical quality, hence when regularly used in animal diets it must be submitted to regular quality control and if necessary, the diet formulation should be adapted in order to avoid variation in animal performance.

In addition to sugars as the main ingredients and a smaller amount of protein, fat, and minerals, soybean molasses is characterized by a significant content of soybean phytochemicals labeled as “soy nutraceuticals”. Soybean molasses contains a wide range of phytochemicals, and the concentration of soybean phytochemicals can be five times higher in soy molasses than in soybeans [20]. The main components of soybean molasses phytochemicals are: isoflavones, saponins, phenolic acids, phospholipids, phytosterols, phytates, ω-3 fatty acids, leucoanthocyanins, Bowman-Birk proteolytic enzyme inhibitors (BBI) [23,29]. Although some soybean phytochemicals showed deleterious effects in animals [30,31] they have a highly beneficial and synergetic effect by delivering antimicrobial, antifungal, antiviral, anti-inflammatory, and antioxidant properties [21]. These attributes of soybean phytochemicals are considered to be largely influenced due to the presence of isoflavones in soybean [32].

### 3.1. Saponins

Saponins are a highly diverse group of triterpene glycosides in which hydrophobic aglycone that may be either a sterol or a triterpene is attached to hydrophilic sugar moieties (pentoses, hexoses, or uronic acids). Saponins are bitter-tasting alcohol-soluble, heat-stable, and amphiphilic compounds [20,33]. Soybean saponins can be divided into three groups based on the chemical structure of the aglycone, the A, B, and E saponins. Hosny and Rosazza [34] isolated and identified soyasaponin A2, soyasaponin I (B saponin), and a new saponin hexaglycoside IV (B saponin) from soybean molasses. Other researchers reported that the ethanol extract prepared from soybean molasses suppressed induced genomic DNA damage, clastogenic damage, and point mutation in mammalian cells [35]. This soybean molasses extract was found to consist of a mixture of group B soyasaponins and 2,3-dihydro-2,5-dihydroxy-6-methyl-4H-pyran-4-one DDMP soybean saponins, which included soybean saponins I, II, III, IV, V, Be, βa, βg, γa and γg (B saponins). Purified soyasapogenol B aglycone isolated from the soybean molasses fraction expressed strong repression in the mutagenic activity in mammalian cells [36]. Soybean saponins were considered in the past to be deleterious antinutritional factors, with little or no justification [37]. The harmful effect of soybean saponins was ascribed by analogy to saponins from other sources that have a toxic effect [38]. Some studies have revealed that saponins are potential functional compounds because they exhibit various biological properties [39]. Generally, there is no consensus on the effects of the application of saponins in animal diets, because these compounds can affect animals in different ways both positive and negative. They have been found to have antioxidative, antifungal, and antiviral effect on one hand, and to significantly reduce the digestion of protein and the uptake of minerals and vitamins in the gut on the other hand [39]. Furthermore, some of the beneficial effects attributed to saponins are positive effects on growth and feed utilization in ruminants and monogastric animals [39]. It was reported that soy-derived saponins can be used as health-promoting feed additives and has a great potential as immunomodulatory compounds that may contribute to sustainable and efficient pork production [38]. The effects of soybean saponins in fish trials are diverse and vary greatly depending on fish species, developmental stage, combination of plant ingredients in the diet, dosage, and duration of exposure [40]. Saponins from soybean molasses showed negative effects on nutrient digestibility and growth and caused enteritis like alterations in the distal intestine in Atlantic salmon [30,31]. On the other hand, sea bream in the grow-out phase demonstrated an ability to tolerate soybean saponins included in the diet within the tested concentration range [40].

### 3.2. Isoflavones

Soybean molasses represents an inexpensive and rich source of isoflavones [41]. Isoflavones are a class of secondary metabolites with a polyphenol structure known as phytoestrogens formed during the growth of a plant [42]. Isoflavones from soybean molasses demonstrated various biological functions, such as anti-inflammatory, antioxidative, and antiproliferative properties [38]. Isoflavones are the source of the bitter and astringent taste in soybean molasses. Soybean contains a mixture of isoflavone compounds based on three “families”: the genistein family, the daidzein family, and the glycitein family occurring in free phenolic or aglycone form. They may also exist in glucoside (daidzin, genistin, and glycitin), acetyl (6-O-acetyldaidzin, 6-O-acetylgenistin, 6-O-acetylglycitin), or malonyl form (6-O-malonyldaidzin, 6-O-malonylgenistin, 6-O-malonylglycitin) [41]. Hosny and Rosazza [34] isolated seven known isoflavones in soybean molasses, genistein, daidzein, glycitein, formononetin, genistin, daidzin, and glycitein 7-O-β-D-6′′-O-acetylglucopyranoside. Soybean isoflavones showed to have variable effects on animal growth, feed conversion ratio, and carcass traits. These effects highly depend on animal species, gender, endocrine status, supplementation level, and timing of exposure [43]. It was found that dietary inclusion of soybean isoflavones increased growth rate and meat deposition in weanling pigs [44], while in another study isoflavones did not have a significant effect on the growth performance of pigs and carcass quality [43]. Soybean isoflavones can be used as feed supplements to decrease fat deposition and increase lean mass in animals due to their estrogen-like function [45]. Furthermore, isoflavones were reported to have significant effects on improving the immune function, reproductive performance, and milk quality of cows [46]. Isoflavones also show good potential as an antioxidant in male broilers because isoflavones can enhance the antioxidative status of male broilers by elevating the activity of antioxidant enzymes [47]. The addition of soybean isoflavones into the diet for laying Japanese quails may improve egg quality and egg production in quail reared under heat stress conditions during the late laying period [48]. However, the deleterious influence of soybean isoflavones on growth performance, bone formation, intestinal function, and liver metabolism in fish was reported [49].

### 3.3. Phenolic Acids

Zhong and Zhao found that soybean molasses had 7.5 times more total phenolic content than other soybean processing by-products such as soybean hull and almost 20 times higher than soybean okara [50]. Soybean molasses contains the following phenolic acid: salicylic acid, gentisic acid, vanillic acid, syringic acid, p-coumaric acid, cinnamic acid, chlorogenic acid, isochlorogenic acid, and ferulic acid [20]. Hosny and Rosazza [34] isolated ferulic acid and two cinnamic acid ester glycosides III from soybean molasses.

### 3.4. Bowman-Birk Inhibitor

The soybean molasses contains approximately 0.2% to 0.5% of the Bowman-Birk trypsin and chymotrypsin inhibitor (BBI). By binding to the digestive enzyme trypsin, BBI had a harmful effect on growth and in some animals could influence pancreatic hypertrophy [37]. On the other hand, much literature has indicated anticarcinogenic effects of BBI by preventing or reducing various types of induced malignant transformations of cells in animals [51].

### 3.5. Fatty Acids

The major saturated fatty acid in soybean molasses is palmitic acid accounting for 11.2–21.2% of the total fatty acid followed by stearic acid, while oleic acid is the most dominant monounsaturated fatty acid in soybean molasses [28,50]. Polyunsaturated fatty acids have the highest share in total fatty acids in molasses (61.2–63.4%), with linoleic acid being the most dominant fatty acid (54–56%). α-linolenic acid was also found in soybean molasses accounting for 7.2% of the total fatty acids [28,50]. The fatty acid content in soybean molasses is shown in Table 2.

### 3.6. Amino Acids

The amino acid profile of soybean molasses is presented in Table 3. According to Zhong and Zhao [50], soybean molasses contains the lowest concentration of total amino acids in comparison to other soybean by-products (e.g., soybean hulls and okara). The most abundant amino acid in soybean molasses is glutamic acid. Soybean molasses was also reported to contain considerable concentrations of arginine, phenylalanine, and aspartic acid [28,50].

## 4. Soybean Molasses in Animal Feed Production

Soybean molasses can be used as a feed ingredient in animal feed production:as a pelleting aidas a source of energy in animal nutrition.

Soybean molasses can be sprayed on soybean meal in a desolventizer-toaster, or can be mixed with soybean hulls and included into liquid rations for animal feeding [20].

### 4.1. Soybean Molasses as a Pelleting Aid

Pelleting is a commonly used processing technique used to transform mash feed into cylindrically shaped macro particles called pellets. Formation of pellets is achieved by pressing the mash through a hole in a metal plate of varying thickness at the conditions of shear and increased moisture and temperature. This processing technique is widely used in the contemporary feed industry. Pelleted feed represents a large part of the estimated 1 billion tons of feed that is produced per year worldwide [52]. Since the second half of the twentieth century, the pelleted diets for poultry dominate over the mash formed diets [53]. Moreover, pellet press is typically used for compacting compound feed for ruminants [54]. The driving force for pelleting is the fact that pellets have superior technical properties over mash, important in handling pellets. The pellets possess better flow properties and higher bulk density that are important in conveying and transport operations [55]. Furthermore, the nutritional quality of feed is enhanced since segregation of the diet ingredients is eliminated in pellets and ingredients are of higher digestibility due to the thermic process which results in gelatinization of starch and denaturation of protein.

The increase of temperature and moisture content of mash feed is crucial before pellet press in order to obtain a product with proper physical quality. The process of conditioning, in which the particles of feed are subjected to mixing while moisture and heat are applied through the addition of steam and water, is therefore paramount for final quality of pellets. Feed pellets are agglomerated products in which ingredients of various particle sizes and water are bound together [56]. The binding between particles can occur through solid bridges, where molecules of different particles come in direct contact and interact. This occurs when pressure is applied during pelleting and the distance between particles is decreasing [55]. The binding is also possible through liquid bonds or better defined as liquid bridges. For the formation of liquid bridges, water is essential, thus liquid bridges are the reason why the addition of water in the conditioning process is so important. Besides water, viscous liquids such as molasses may contribute to the formation of liquid bridges. Upon cooling of pellets water from liquid bridges evaporates, leaving dry particles that form solid bridges and create strong bond [55]. If molasses is present in the pelleted diet, then the cooling process results in a glass-like formation, which has strong binding properties. This formation occurs due to recrystallization of sugar from molasses when water is removed [57]. The solubilized sugar presented in molasses may have positive effects on pellet hardness and durability. Therefore, adding molasses in mixer or in a conditioner together with steam may aid compaction of particles during the pelleting process [52]. In the study by Dunmire [58], liquid sugar cane molasses was used as a substitute for whey permeate in nursery pig diets at a level of 9.4% significantly improved pellet durability and reduced amount of fines after the pelleting process. As far as the authors of this review are aware, there is no literature that investigates the influence of addition of soybean molasses in animal feed diets on the pelleting process and pellet quality. The authors assume that the high sugar content present in soybean molasses would act as a binder of particles if present in pelleted diets. As it is reviewed in a study by Thomas et al. [55], the addition of cane and beet molasses in diets provided not just more durable pellets but improved specific capacity of pellet press. Therefore, the assumption is that soybean molasses would also have the same effects on pelleting process and physical quality of obtained pellets.

### 4.2. Soybean Molasses as a Feed Ingredient

#### 4.2.1. Use of Soybean Molasses in Ruminant Diets

Soybean molasses can be used in making silage, by preserving, packing, and compacting it in a bunker or silo. Under these conditions, the living cells quickly consume the oxygen entrapped in the mass and release carbon dioxide, which prevents the growth of moulds. Moreover, the addition of molasses provides a source of sugars for the acid formic bacteria which generally improve the value of silage [59]. Soybean molasses showed a good potential to be used in liquid feed diets for ruminants because it provides multiple beneficial effects on animals. Firstly, it can reduce dustiness of concentrates and total mixed rations and prevent selective consumption of diet compounds, thereby preventing the occurrence of acidosis [60]. Soybean molasses can be used as a lick to stimulate appetite and condition of the animals. Because of its sweet taste, soybean molasses enhances the taste and overall acceptance of coarse and less tasty bulky feedstuff, poor quality damaged hay or pastures, and thus leads to increased consumption of dry matter of meal [61]. Additionally, in ruminants, soybean molasses might affect growth and performance increasing, mainly ruminal fiber digestibility that leads to greater dry matter intake [60,61,62,63]. These effects have been reported in sheep fed basic diet supplemented with 2%, 5%, and 8% soybean molasses [64,65]. Schultz and co-workers [66] evaluated the effect of sheep supplementation with soybean molasses on feed intake, digestibility, feeding behavior and metabolic profile. These results have been confirmed in subsequent studies, in which soybean molasses was added to a basal corn silage diet at increasing concentration levels (0, 3, 6, 9%) up to 12% in the dry matter. It was found that crude protein intake increased proportionally with the increase in soybean molasses, which, therefore, was a protein delivery medium in the diet. Moreover, feed digestibility, feed consumption behavior, and metabolic profile of sheep were not affected, indicating that soybean molasses can be used successfully as a feed ingredient for sheep until 12% of the daily ration without any risk of intoxication. The influence of soybean molasses on the ruminal environment has been also investigated [67,68,69,70,71]. Sheep diet containing 0%, 15%, or 30% soybean molasses (dry matter basis) increased proportionally ruminal pH and decreased NH3-N concentration, thereby improving the ruminal fermentation. Further effects have been reported in greenhouse gas production in feedlot sheep: the use of soybean molasses up to 20% linearly decreased total gas and CH4 production and improved feed efficiency [68]. 

Soybean molasses has been sought as a valuable feed ingredient in the diet for dairy cows concerning the fact that it contains a significant amount of carbohydrates, which are the most important source of energy in dairy cows’ diets accounting for 60–70% of the total dry matter of the diet. The addition of soy molasses to the diet for dairy cows increases the rate of carbohydrates, whose degradation provides energy for the development of ruminal microorganisms and the host animal [69]. This leads to a higher intensity of microbial protein synthesis from the available non-protein nitrogen. Sugars from soybean molasses can improve the efficiency of rumen nitrogen utilization [70], reduce the concentration of ammonia nitrogen in the liquid content of rumen, and increase milk protein yield [71]. Adequate use (<15% per dry matterDM) of soybean molasses in the dairy cows feeding can also affect the increase in butyrate production in rumen, which enhances blood flow through the rumen epithelium. It may result in a more rapid transfer of volatile fatty acids from epithelial cells to the bloodstream and an increase in the ruminal pH value [72]. It was reported that soybean molasses can be successfully used in consumed total mixed rations for lactating dairy cows as a source of fast fermentable carbohydrates in rumen, without any impact on rumen pH [73]. 

Soybean molasses was compared to soybean meal and included as a feed ingredient to finishing diets for steers which contained either flaked corn or a combination of high-moisture corn and dry-rolled corn [74,75,76]. Thus, in beef cattle, it has been found that soybean molasses can increase feed intakes and daily gain and the main reason for that has been attributed to its proteins and energy contents. The influence of soybean molasses on in vitro ruminal fermentation of the finishing diet for steers was also evaluated [75]. Combining these results, it can be suggested that diets containing soybean molasses up to 15% can be used in ruminant species without negative effects on ruminal fermentation and performance.

#### 4.2.2. Use of Soybean Molasses in Pig Diets

Although there are no papers dealing with the use of soybean molasses in feed for pigs, this alternative feed ingredient has a promising potential to be included in the pig diet as an energy source since pigs are able to digest the oligosaccharides present in soybean molasses. Stachyose and raffinose being highly distributed in soybean molasses were completely fermented by the hindgut bacteria of the weanling pig [77]. The incorporation of soybean molasses at a certain level in feed has the potential to improve growth performance and alleviate the adverse effect of heat stress in growing pigs [78]. Molasses has a laxative effect on animals when used at high levels in diets for pigs, especially those for young pigs may cause diarrhea [79]. The digestive disturbances and laxative effect of molasses are often attributed to a high concentration of potassium ions present in the molasses, which may disturb electrolyte balance and increase osmotic pressure [80]. Therefore, it is expected that soybean molasses could be included in feed for young pigs at a lower level, while adult or near-adult pigs could tolerate a higher level of molasses incorporation.

#### 4.2.3. Use of Soybean Molasses in Poultry Diets

There are no published studies on the use of soybean molasses in poultry nutrition to date. However, soybean molasses has the potential to be added to the diet for poultry but at a lower level compared to diets for ruminants and pigs. Molasses is commercially included in poultry rations at the level of 5%. The use of soybean molasses is limited because higher inclusion levels can cause toxicity and laxative effects in chickens [81]. However, the inclusion of soybean molasses in a smaller amount could improve feed consumption and animal performance in broilers and layers [81].

#### 4.2.4. Use of Soybean Molasses in Fish Diets

The inclusion of soybean molasses into aquaculture nutrition showed to be restrictive because higher inclusion levels (>10%) had a detrimental effect on nutrient digestibility and caused morphological changes in the distal intestine of salmon. Namely, Olli and Krogdahl [30] studied the effect of the addition of soybean molasses at different levels (5%, 10%, 15%, and 20%) in a fish meal-based diet for Atlantic salmon (Salmo salar L.) on nutrient digestibility. It was found that the digestibility of fat was significantly reduced from 86% to 76% with increasing the inclusion level (>10%) of soybean molasses, whereas digestibility of protein and dry matter were not significantly affected. The inclusion of soybean molasses into the salmon diet had a more severe effect on the digestibility of saturated- and monounsaturated fatty acids than those of polyunsaturated fatty acids. In fact, the digestibility of C16:0 fatty acid was influenced at the inclusion level >10%, and of C20:l and C22:l when soybean molasses was added at the level >15%. At the inclusion level of 20%, the digestibility of C20:l and 22:1 was reduced by approximately 20% and 30%, respectively. No changes in the digestibility of C20:5 and C22:6 were observed with the addition of soybean molasses. The effects of soybean molasses were dose-dependent which indicated that lower inclusion concentration was preferable in order to not deliver negative effects on fish. In another study, the inclusion of soybean molasses in the diet for Atlantic salmon caused the development of morphological alterations in the distal intestine [82]. In the experiment, salmon weighing 145 g were kept in seawater and fed diet containing 15% of soybean molasses. The morphological changes observed in the distal intestine in salmon were reflected in a significant decrease of enterocytes or absence of absorptive vacuoles and enlarged and hypercellular stroma. The influence of soybean molasses inclusion in the diet for salmon on enteritis-like alterations in the distal intestine was also confirmed by Krogdahl et al. [31]. In that study, soybean molasses caused increased levels of both lysozyme and IgM measured in the mid and distal intestinal mucosa, indicating inflammatory response that could cause increased susceptibility to furunculosis.

Soybean molasses contains components that might affect the digestibility of fat, cause enteritis in the distal intestine of salmon, and show the same signs of inflammation as fish fed soybean meal. Olli and Krogdahl [30] suggested that saponins from molasses may interact with cholesterol and with bile salt, which is necessary for fat digestion and consequently reduced fat digestibility. In order to trace the causative components for soybean molasses-induced enteritis in Atlantic salmon, Knudsen et al. [83] separated soybean molasses into three sub-fractions (butanol phase, precipitate, and water phase) using water-saturated n-butanol. Fish fed soybean molasses exhibited the same alterations in histological parameters (vacuoles, lamina propria, connective tissue, and mucosal folds) as fish fed combination of butanol phase and precipitate. The authors discovered that saponins were causative compounds for enteritis in salmon because they were the only quantified compounds being split between the precipitate and the butanol phase. It was suggested that saponins alone or saponins in combination with the gut microflora or antigenic soybean proteins triggered the inflammatory reaction in the distal intestine of salmon. Moreover, soybean molasses still induced enteritis after being processed with n-butanol at 70°C for more than 1 h, which indicated that causative components in molasses could withstand alcohol treatment at elevated temperatures.

Even though research indicated that saponins from soybean molasses can cause soybean-induced enteritis in salmonids, Krogdahl and Bakke [33] concluded that saponins when pure compound will not cause enteritis in a concentration below 2 g/kg unless mixed with other plant materials (e.g., lupin kernel meal or pea protein concentrate) in the diet. Therefore, cautions should be taken when mixing plant ingredients containing saponins, especially legumes, in diets for farmed fish. According to Francis [84], the concentration of saponins below 1 g/kg of diet is considered safe and unlikely to affect the growth performance of common culture fish.

## 5. Limitations of Commercial Application of Soybean Molasses in Animal Nutrition

The major challenge for the application of soybean molasses is its form of storage and delivery. It is mostly stored in bulk containers and that presents the issue for some small animal feed producers that lack storage space in their mills. Relatively short shelf-life of soybean molasses (6-12 months according to manufacturers) may also pose an issue for the feed producers when it is not regularly used in diets [16]. Just like other liquid feed ingredients, soybean molasses has to be well mixed with other feed compounds in order to reduce selective consumption by animals, feed intake, and occasional decrease in rumen pH [16]. The utilization of soybean molasses in animal feeding might be limited because overconsumption of molasses could cause laxative effects in monogastric animals. However, soybean molasses has a lower viscosity than other molasses which are obtained as a by-product of the sugar industry (beet and cane molasses), which significantly facilitates its manipulation and application in animal feed especially during the winter months [73]. Even so, soybean molasses is still a viscous and sticky liquid that must be uniformly added into the mixer during feed production by spraying through the nozzle. Otherwise, it can create lumps in mash feed which can negatively influence diet homogeneity or pellet quality, and it also requires frequent cleaning since it can create build-ups in mixer or pellet die and, therefore, increase wear of the equipment [58].

## 6. Conclusions

The reduction of biomass waste from the agro-food industry and its efficient valorization is a key factor contributing to the sustainability of the livestock sector. The knowledge about soybean molasses valorization in animal diets has notably increased in recent years. In animal nutrition, soybean molasses may serve as a pelleting aid during feed production. Additionally, it is considered a rich source of energy and phytochemicals, and its inclusion into animal diets not only can ameliorate the feed value and quality but also can improve animal performance and health. Soybean molasses is still by far one of the cheapest feed ingredients available to farmers, which justifies its application on large scale as animal feed, especially for ruminants. Based on the literature, soybean molasses has the greatest potential to be used in diets for ruminants, while up to now there has been no documented research on soybean molasses application in diets for monogastric animals regardless of the fact that it could be included at smaller levels. Taking into account the many benefits of using soybean molasses in animal nutrition, further research of this alternative feed ingredient is needed to expand and stimulate its efficient use in a clean and sustainable way.

## Figures and Tables

**Figure 1 animals-11-00514-f001:**
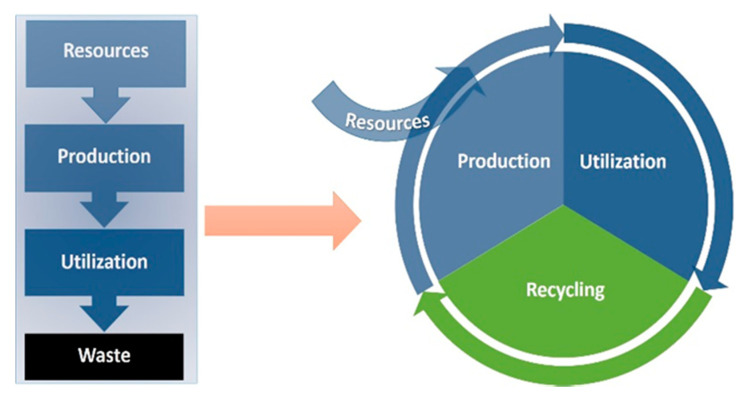
From a linear to a circular agro-food system (Source [1]).

**Figure 2 animals-11-00514-f002:**
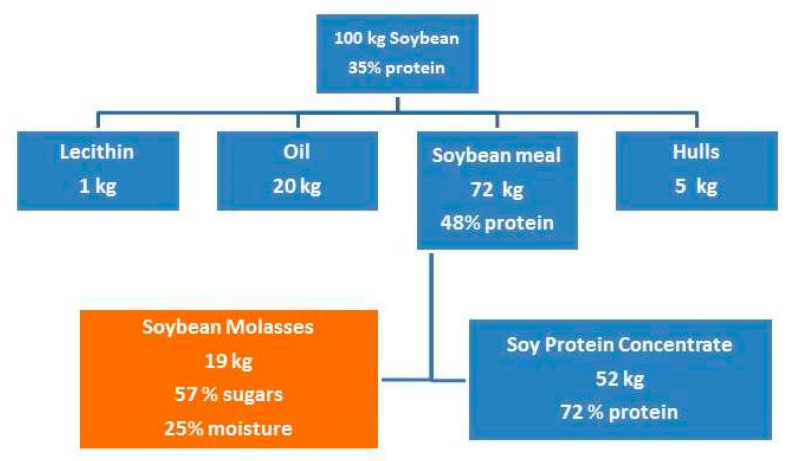
Products of the industrial processing of soybean (Source [22])**.**

**Table 1 animals-11-00514-t001:** Proximate chemical composition of soybean molasses [17,18,23,24,25,26,27].

Component	Content (% d.m.)
Crude matter	53.1–82.5
Crude protein	5–12
Crude fat	4–20
Ash	3–21.9
Sugars	58–65
Oligosaccharides	
Stachyose	15.5–34.2
Raffinose	4–25.5
Disaccharides	
Saccharose	18.5–32
Monosaccharides	
Fructose	1.2–3.0
Glucose	0.2–4.7
Saponins	6–15
Isoflavones	0.8–2.5
Other compounds (phenolic acids, leucoanthocyanins, etc.)	difference to 100
Total energy	1400 kJ/100g

**Table 2 animals-11-00514-t002:** Fatty acid profile of soybean molasses [28,50].

Fatty Acids	Content (%)
Myristic acid (C14:0)	0.1–0.2
Pentadecanoic acid (C15:0)	0.1
Palmitic acid (C16:0)	11.2–21.2
Palmitoleic acid (C16:1)	0.1
Margaric acid (C17:0)	0.2
Stearic acid (C18:0)	3.8–3.9
Trans oleic acid (C18:1n9t)	0.1
Oleic acid (C18:1n9c)	9.9–23.1
Trans linoleic acid (C18:2n6t)	0.2
Linoleic acid (C18:2n6c)	54–56
Arachidic acid (C20:0)	0.1–0.3
Eicosenoic acid (C20:1)	0–0.2
α-linolenic acid (C18:3n3)	7.2
Behenic acid (C22:0)	0.4–0.5
Tricosylic acid (C23:0)	0.2
Erucic acid (C22:1)	0.2
Lignoceric acid (C24:0)	0.2–0.3
Saturated fatty acids	16.2–26.4
Monounsaturated fatty acids	10.1–23.6
Polyunsaturated fatty acids	61.2–63.4

**Table 3 animals-11-00514-t003:** Amino acid profile of soybean molasses [28,50].

Amino Acids	Content (g/kg Dry Matter)
Aspartic acid	3.2–8.7
Threonine	1.8–2.9
Serine	1.3–3.6
Glutamic acid	5.0–13.7
Proline	2.2–3.8
Glycine	0.9–3.2
Alanine	2.2–3.4
Cystine	1.2–7.6
Valine	1.6–3.7
Methionine	1.1–3.6
Isoleucine	1.5–3.5
Leucine	1.2–5.9
Tyrosine	2.7–4.6
Phenylalanine	3.9–5.1
Histidine	2.1–5.4
Lysine	0.9–4.8
Arginine	5.6–7.9

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
