# Peer review of "Soybean Molasses in Animal Nutrition"

_animals, 2021, doi:10.3390/ani11020514_

Round 1

Reviewer 1 Report

Interesting review article, which incorporates production, composition and use in animal production.

Despite the difference in research carried out on its use, i suggest simplifying point 4.2.1.

References must be revised, standardizing their presentation (dates and formats)

In the tables, rounding numbers must be standardized (Tenths)

Body of the attached article includes comments

Author Response

Reviewer 1

Dear Reviewer thanks for your comments.

Specifically:

L52: thanks for your suggestion. Reference 5 has been changed with a more recent one.

L96: as suggested by reviewer the reference has been changed

L501-517 the reference has been adapted.

Reviewer 2 Report

Well written review paper on a critical aspect of biomass waste, specifically soybean molasses, and it potential as a component of animal diets.

Specific suggestions/edits:

Page      Line

1            12         insert ‘the’ after of

1            15         diets

1            16         insert ‘a’ after is  

1            21         biomass waste (change here and throughout manuscript)

1            23         replace imposed with ‘proposed’

1            29         Start sentence with ‘Special…’

2            51-52    Suggest rewriting this sentence – reads awkwardly

2            53         ingredients

2            68         insert ‘a’ after generate

2            75         contributes

2            72-73    include citation here

2            80         diets

3            88         does the choice of alcohol used in this process alter the                                  chemical/nutritional value of the molasses? Authors should                               address.

3            98         spell out approximately

4            117-135 this section illustrates the wide range of chemical                             composition of soybean molasses. Authors should discuss and bring                 this forward to readers. This could potentially have a major impact on               its nutritional value, ultimately impacting animal performance.

4            146       phytochemicals

5            155       divided

5            170       authors should suggest potential impacts of saponins on                                  other animal diets

5            172       second part of this sentence read awkwardly – suggest                                     rewrite not using ‘concerning’

5            174       insert ‘a’ of

5            172-190  again, I think it is good to briefly include some of the                                     impacts of the chemical composition of molasses on human                             health, but the discussion should include more animal diet                               discussion in a review manuscript

6            205-207 include citation here

7            230 insert ‘a’ after is     

7            230-232 suggest a rewrite this lengthy sentence

7            246       delete ‘are’ following sizes

8            261-263 suggest including citations and discussion by authors are                                 their assumption

8            264-274 – suggest deleting this section – redundant information

 8           281       animals

9            299       remove p values here and throughout – ‘significantly’                                      comminutes that to the reader  

9            327       replace neither with either and nor with or

9            334       insert comma after ‘When studied,..’

9            338       replace indexes with indices

9            342       change to ‘level of propionic acid was increased.’

Author Response

Reviewer 2

Dear Reviewer thanks for your comments. Here below our answers.

Page      Line

1            12         insert ‘the’ after of                          done

1            15         diets                                                done

1            16         insert ‘a’ after is                             done

1            21         biomass waste (change here and throughout manuscript) done

1            23         replace imposed with ‘proposed’   done

1            29         Start sentence with ‘Special…’      done

2            51-52    Suggest rewriting this sentence – reads awkwardly done

2            53         ingredients                                      done

2            68         insert ‘a’ after generate                  done

2            75         contributes                                      done

2            72-73    include citation here                       done

2            80         diets                                                done

3            88         does the choice of alcohol used in this process alter the                                  chemical/nutritional value of the molasses? Authors should                               address. As suggested by the reviewer, the authors clarified the connection between the choice of alcohol used and chemical/nutritional value of the molasses (L91-92).

3            98         spell out approximately                  done

4            117-135 this section illustrates the wide range of chemical                             composition of soybean molasses. Authors should discuss and bring                 this forward to readers. This could potentially have a major impact on               its nutritional value, ultimately impacting animal performance. As suggested by the reviewer, the authors discussed the influence of wide range of chemical composition of soybean molasses on nutritional value and animal performance (L138-141).

4            146       phytochemicals                               done

5            155       divided                                            done

5            170       authors should suggest potential impacts of saponins on                                  other animal diets As suggested by the reviewer, the authors revised that part and included impacts of saponins on other animal diets  in lines L174-191.

5            172       second part of this sentence read awkwardly – suggest                                     rewrite not using ‘concerning’             done

5            174       insert ‘a’ of                                                                        done

5            172-190  again, I think it is good to briefly include some of the                                     impacts of the chemical composition of molasses on human                             health, but the discussion should include more animal diet                               discussion in a review manuscript As suggested by the reviewer, the authors thoroughly revised that part and included the effect of soy isoflavons in animal diet in lines 209-223.

6            205-207 include citation here                                             done

7            230 insert ‘a’ after is                                                           done  

7            230-232 suggest a rewrite this lengthy sentence. The sentence was revised as suggested.

7            246       delete ‘are’ following sizes                                    done

8            261-263 suggest including citations and discussion by authors are  their assumption 

reference added

8            264-274 – suggest deleting this section – redundant information As suggested by the reviewer, the section in lines 264-274 was deleted.

 8           281       animals                                           done

9            299       remove p values here and throughout ‘significantly’    comminutes that to the reader                                       done

9            327       replace neither with either and nor with or                                               done

9            334       insert comma after ‘When studied,..’                                            done

9            338       replace indexes with indices                                              done

9            342       change to ‘level of propionic acid was increased.’           done

Reviewer 3 Report

Comments to the Authors:

This is an interesting review article about the use of soybean molasses, as a by-product of the soybean processing industry, in animal nutrition. Increasing sustainability is currently one of the greatest challenges in animal production and the potential utilization of this by-product is a way to help. So the topic is relevant in animal nutrition.

The article is well structured, although it should be improved to make it more understandable. Specific comments to help are as follows:

L 51-52: The statement here seems wrong or unclear. You have mentioned that livestock production is rapidly growing, so it seems odd to state that the global demand for animal products in developing countries will be reduced (more than half) by 2030.

L 142-148: As explained, there are several different phytochemicals in soybean molasses, and I would think that not all of them have all the beneficial properties stated here. Please, make this clear.

L 152-163: This paragraph is a bit confusing. You state that, based on the chemical structure, soybean saponins can be divided into three groups, although then you mention several saponins and do not mention which group they belong to.

L 179-190: As previously mentioned with saponins, the explanation on soybean isoflavones is a bit confusing. I suggest to describe first the groups or families of isoflavones and then report which is the type of isoflavone you mention. Is it glycitin or glycitein? I assume that the properties described in lines 186-188 for soybean molasses are due to the isoflavones, although it is not clear enough.

L 194: This is the first time you mention the soybean okara, which is not included in figure 2. Do you think it may be worth adding this by-product in the figure?

Table 2: Is there only one article reporting the fatty acid profile of molasses? Please, correct “monosaturated” and “polysaturated” to “monounsaturated” and “polyunsaturated” fatty acids.

Table 3: Please, give the references used for this table.

L 288: What do you mean with “undegraded intake protein”? It is not clear.

L 305-308: I agree that the addition of soybean molasses can stimulate the production of volatile fatty acids and decrease the pH, although, depending on the environment, this effect may not be beneficial in the rumen. This contrasts with the ideas explained in lines 332 (where it is mentioned a study where the addition of soybean molasses increased ruminal pH) and 370. Please, discuss further and reword some parts to make clearer this issue. In general, I think that this section (“Use of soybean molasses in ruminant diets”) should be revised and modified to explain the ideas clearer. It seems that you have tried to explain separately the effects on different species of ruminants (sheep and cattle) or physiological status (dairy cows and steers), although it would be good to discuss, at least briefly in some parts, the results referring to similar parameters (such as pH or total volatile fatty acids).   

L 428-429: Please, give a reference to support the statement that high levels of inclusion of soybean molasses can cause toxicity and laxative effects in chickens.

L 433: You state that “higher inclusion levels had detrimental effect”, although there is no clear comparison (higher inclusion levels than what?).

L 438: Do you mean “digestibility of protein and dry matter were not significantly affected”? Please, revise and correct.

L 438-439: Please, revise and specify whether long-chain refers to saturated and monounsaturated fatty acids. It is unclear.

L 479-480: This sentence is unclear, so, please, reword it to make it clearer.

L 485-486: This explanation of the laxative effects should be also included the first time you mention this impact (in the section of “Use of soybean molasses in pig diets”).

The English writing should be thoroughly revised along the manuscript since in some parts it precludes a clear understanding of the text.

Author Response

Reviewer 3

Dear Reviewer thanks for your comments. here belo our answers.

Comments to the Authors:

This is an interesting review article about the use of soybean molasses, as a by-product of the soybean processing industry, in animal nutrition. Increasing sustainability is currently one of the greatest challenges in animal production and the potential utilization of this by-product is a way to help. So the topic is relevant in animal nutrition.

The article is well structured, although it should be improved to make it more understandable. Specific comments to help are as follows:

L 51-52: The statement here seems wrong or unclear. You have mentioned that livestock production is rapidly growing, so it seems odd to state that the global demand for animal products in developing countries will be reduced (more than half) by 2030.

the refence has been reviewed

L 142-148: As explained, there are several different phytochemicals in soybean molasses, and I would think that not all of them have all the beneficial properties stated here. Please, make this clear. As suggested by the reviewer, the part in lines 142-148 was revised and better explained (L148-151). 

L 152-163: This paragraph is a bit confusing. You state that, based on the chemical structure, soybean saponins can be divided into three groups, although then you mention several saponins and do not mention which group they belong to. The authors appreciate the reviewer comment and marked the group soybean molasses saponins belong to according to the chemical structure.

L 179-190: As previously mentioned with saponins, the explanation on soybean isoflavones is a bit confusing. I suggest to describe first the groups or families of isoflavones and then report which is the type of isoflavone you mention. Is it glycitin or glycitein? I assume that the properties described in lines 186-188 for soybean molasses are due to the isoflavones, although it is not clear enough. The authors appreciate the reviewer comment and described first the groups or families of isoflavones and then reported the type of the mentioned isoflavones. Glycitein occurs in aglycone form, while glycitin occurs in glycosides form.  Also, the sentence in lines 186-188 was checked and omitted from the manuscript.

L 194: This is the first time you mention the soybean okara, which is not included in figure 2. Do you think it may be worth adding this by-product in the figure? Soybean okara is a by-product of the production process of soybean food. Precisely, it is the residue of soybean milling after extraction of the aqueous fraction used for producing tofu and soydrink (doi: 10.1016/j.lwt.2018.02.058). Therefore, it was not included in Figure 2.

Table 2: Is there only one article reporting the fatty acid profile of molasses? Please, correct “monosaturated” and “polysaturated” to “monounsaturated” and “polyunsaturated” fatty acids.  The authors found another paper reporting the fatty acid profile of soybean molasses and included data in Table 2.

Table 3: Please, give the references used for this table.

L 288: What do you mean with “undegraded intake protein”? It is not clear. Rumen undegraded protein (RUP), or commonly referred as bypass protein, is the fraction of crude protein consumed by ruminants which is not degraded by rumen microbes. Increase in rumen undegraded protein (RUP) of protein feedstuff improves the efficiency of utilization of ruminal ammonia N for milk protein synthesis in dairy cows and also results in manure with significantly lower ammonia-emitting potential.

L 305-308: I agree that the addition of soybean molasses can stimulate the production of volatile fatty acids and decrease the pH, although, depending on the environment, this effect may not be beneficial in the rumen. This contrasts with the ideas explained in lines 332 (where it is mentioned a study where the addition of soybean molasses increased ruminal pH) and 370. Please, discuss further and reword some parts to make clearer this issue. In general, I think that this section (“Use of soybean molasses in ruminant diets”) should be revised and modified to explain the ideas clearer. It seems that you have tried to explain separately the effects on different species of ruminants (sheep and cattle) or physiological status (dairy cows and steers), although it would be good to discuss, at least briefly in some parts, the results referring to similar parameters (such as pH or total volatile fatty acids).   

the paragraph has been simplified (as requested by two reviewers) and deeply revised (L304-352).

L 428-429: Please, give a reference to support the statement that high levels of inclusion of soybean molasses can cause toxicity and laxative effects in chickens.

In order to support the statement, the authors included the reference which also refer to other papers explaining the toxic and laxative effect in chickens because of high inclusion levels of molasses (doi: 10.3382/ps.0510821, doi: 10.3382/ps.0510813, doi: 10.1080/713654976).

L 433: You state that “higher inclusion levels had detrimental effect”, although there is no clear comparison (higher inclusion levels than what?).

The authors added the soybean molasses inclusion level.

L 438: Do you mean “digestibility of protein and dry matter were not significantly affected”? Please, revise and correct.

The phrase was revised and corrected as suggested.

L 438-439: Please, revise and specify whether long-chain refers to saturated and monounsaturated fatty acids. It is unclear.

The sentence was revised and corrected as suggested.

L 479-480: This sentence is unclear, so, please, reword it to make it clearer.

The sentence was revised and corrected as suggested.

L 485-486: This explanation of the laxative effects should be also included the first time you mention this impact (in the section of “Use of soybean molasses in pig diets”).

As suggested by the reviewer, the explanation of the laxative effects was included in the section of “Use of soybean molasses in pig diets”.

The English writing should be thoroughly revised along the manuscript since in some parts it precludes a clear understanding of the text.

The English writing was thoroughly revised along the manuscript as suggested.